# Linear-Time Modeling of Linguistic Structure:
# An Order-Theoretic Perspective

**Tianyu Liu**     **Afra Amini**     **Mrinmaya Sachan**     **Ryan Cotterell**

{`tianyu.liu`, `afra.amini`, `mrinmaya.sachan`, `ryan.cotterell`}@inf.ethz.ch

**ETH** *zürich*

## Abstract

Tasks that model the relation between pairs of tokens in a string are a vital part of understanding natural language. Such tasks, in general, require exhaustive pair-wise comparisons of tokens, thus having a quadratic runtime complexity in the length of the string. We show that these exhaustive comparisons can be avoided, and, moreover, the complexity of such tasks can be reduced to linear by casting the relation between tokens as a partial order over the string. Our method predicts real numbers for each token in a string in parallel and sorts the tokens accordingly, resulting in total orders of the tokens in the string. Each total order implies a set of arcs oriented from smaller to greater tokens, sorted by their predicted numbers. The intersection of total orders results in a partial order over the set of tokens in the string, which is then decoded into a directed graph representing the desired linguistic structure. Our experiments on dependency parsing and coreference resolution show that our method achieves state-of-the-art or comparable performance. Moreover, the linear complexity and parallelism of our method double the speed of graph-based coreference resolution models, and bring a 10-times speed-up over graph-based dependency parsers.

○ https://github.com/lyutyuh/partial

## 1 Introduction

Strings of tokens in natural language are not constructed arbitrarily. Indeed, which tokens co-occur within the same string is highly structured according to the rules of the language. Understanding such structures is critical to the comprehension of natural language. In natural language processing (NLP), many structured prediction tasks aim to automatically extract the underlying structure that dictates the relationship between the tokens in a string of text. Examples of such tasks include dependency parsing, semantic parsing, and coreference resolution. These tasks involve predicting complex and hierarchical output structures, making them inherently more challenging than their classification or regression counterparts. This paper contributes a novel and generic framework for structured prediction with empirical evidence from dependency parsing and coreference resolution.

Many machine learning models for structured prediction score and predict graphs (McDonald et al., 2005; McDonald and Pereira, 2006), in which the vertices represent the tokens in the string and the edges represent the relations between them. One common strategy to model a graph is to decompose it into smaller subgraphs that are tractable (Taskar et al., 2004; Smith, 2011, §2.2). For example, arc-factored models (Eisner, 1996) score a graph only using the score of each constituent edge. However, even with such simplification, the computational costs of arc-factored models are superlinear. The reason is that one needs to exhaustively compute scores for all possible edges in the graph, which, in general, requires at least *quadratic* number of computations with respect to the length of the string. Another common strategy employs weighted transition-based systems (Knuth, 1965; Yamada and Matsumoto, 2003; Nivre, 2003). They decompose structures into transitions between intermediate model states and *do* offer linear-time algorithms. However, in general, predicting the transitions between states cannot be parallelized, which is another worrying limitation. The authors of this paper contend the limitations of both graph-based and transition-based models are frustrating in an era when researchers are processing longer and longer texts (Tay et al., 2021).

From a more abstract perspective, the mathematical and algorithmic foundation on which structured prediction models rest can be regarded as a *design choice*. Graph-based and transition-based modeling are both specific design choices. These design

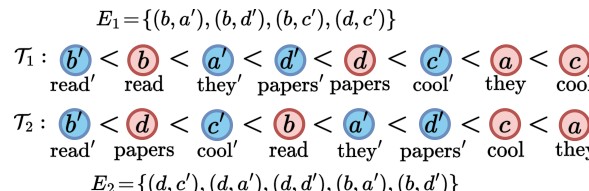

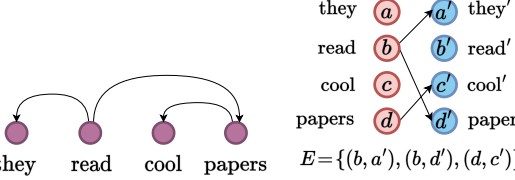

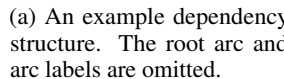
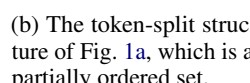

(a) An example dependency structure. The root arc and arc labels are omitted.

(b) The token-split structure of Fig. 1a, which is a partially ordered set.

(c) A realizer of Fig. 1b with 2 total orders such that $E = E_1 \cap E_2$. $E_1$ and $E_2$ contain the arcs oriented from $V^r$ (red nodes) to $V^b$ (blue nodes) and from left to right.

Figure 1: An overview of our method. To model a linguistic structure, represented as a directed graph in Fig. 1a, we first convert it into a token-split structure (see §3.4) in Fig. 1b, which is a partial order, to remove undesired transitivity. Then, 2 real numbers are predicted for each vertex in Fig. 1b. The positions of vertices in Fig. 1c in the inequalities indicate the real numbers the vertices are mapped to. The vertices are sorted twice accordingly, resulting in a realizer (see Def. 3.8) of 2 total orderings, each possessing a set of edges $E_1$ and $E_2$. The exact set of desired edges in the original structure can be restored from the intersection of $E_1$ and $E_2$ (see §3.3). Some qualitative examples are included in App. J.

choices impose substantial *inductive biases* by confining the class of models available to be utilized to solve the task and set limits on the efficiency of the models. In this paper, we propose a fresh design choice for structured prediction. Specifically, we propose an *order-theoretic* perspective to understand and model structures in NLP. Our approach can predict many structures in natural language in $\mathcal{O}(N)$ time where $N$ is the length of the string and is easily parallelizable. The linear-time complexity means our method avoids comparing all $\mathcal{O}(N^2)$ token pairs. The key innovation that enables this speed-up is the following: Rather than considering structures as graphs, we view them as *partial orderings* of the tokens in the strings.

Concretely, we treat structured prediction as a regression task. Because the set of real numbers $\mathbb{R}$ is naturally ordered by $<$, we use real numbers as the proxy for determining the partial order. We predict $K$ numbers for each token and sort the tokens $K$ times accordingly. Two tokens are partially ordered by $\prec$ if and only if they are ordered by $<$ in *all* of the $K$ orders above. We further provide an efficiency guarantee based on the well-established result in order theory that partial orders satisfying particular conditions can be represented as the intersection of as few as $K = 2$ total orders. We show that most structures in natural language, including trees, alignments, and set partitions, satisfy these conditions. This result enables us to develop a linear-time algorithm for predicting such structures. Fig. 1 gives an illustrative example of our framework applied to dependency parsing, in which the structure being modeled is a tree.

On dependency parsing, our experimental results

show that our method achieves 96.1 labeled attachment score (LAS) and 97.1 unlabeled attachment score (UAS) by using an intersection of only 2 total orders, 96.4 LAS and 97.4 UAS using an intersection of 4 total orders on the English Penn Treebank (Marcus et al., 1993). Furthermore, our method sets the new state of the art on Universal Dependencies 2.2 (Nivre et al., 2018), while being 10 times faster and more memory efficient than graph-based models. Our method also achieves 79.2 F1 score with only 4 total orders on the English OntoNotes coreference resolution benchmark (Pradhan et al., 2012), which is on par with the state of the art, while being twice as fast and using less memory.

## 2 Motivation

We now provide high-level motivation for order-theoretic structured prediction.

### 2.1 Linearization of Structure

The NLP literature abounds with linear-time structured prediction models. Many are derived from the classical shift–reduce parsers (Knuth, 1965) from the compiler literature. One recent line of research has derived linear-time parsers by reducing parsing to tagging (Gómez-Rodríguez and Vilares, 2018; Strzyz et al., 2020; Kitaev and Klein, 2020; Amini et al., 2023, *inter alia*). In these methods, a *finite* set of tags $\mathcal{C}$ is chosen such that all structures for parsing a string can be embedded in $\mathcal{C}^N$ for a string of length $N$. Tagging-based parsers often yield strong empirical performance in both constituency parsing and projective dependency parsing. A natural question is, then, why do we need another method?

We give two motivations. The first linguistic

and the second mathematical. Linguistically, the underlying structures of natural language, e.g., syntax, semantics, and discourse, are often *not* aligned with the surface form of a sequence due to the existence of **displacement** (Chomsky, 2015, Chapter 1, p. 44). The strong performance of parsing-as-tagging schemes relies, in part, on there being a tight correspondence between the surface string and structure (Amini and Cotterell, 2022, Proposition 1). Mathematically, the maximum number of structures that a discrete tag sequence can represent is at most $\mathcal{O}(|\mathcal{C}|^N)$. This set is simply not large enough to capture many structures of interest in NLP. For instance, the space of non-projective dependency trees of $N$ tokens has a cardinality of $\mathcal{O}(N^{N-2})$ (Cayley, 1889). Therefore, to parse non-projective dependency trees with tagging, the size of the tag set has to grow with $N$. However, this implies performing a classification task with an *infinite* number of classes.

## 2.2 An Illuminating Example

Order-theoretic approaches appear across computer science. For instance, it is well-known that a binary tree can be uniquely restored from its inorder traversal and either the pre- or postorder traversal. Consider the following binary tree.

**Example 2.1** (Binary Tree)**.**

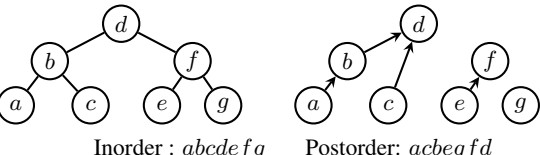

Inorder : $abcdefg$  Postorder: $acbegfd$

Figure 2: An example binary tree and a partial order over the vertices induced by two total orders. ∎

In a binary tree, a vertex $x$ is a left descendant of vertex $y$ *if and only if* $x$ is visited before $y$ in both of the in- and postorder traversal. E.g., in Ex. 2.1, $a$ is the left descendant of $d$ and is visited before $d$ in both the in- and postorder traversal.

Another way of stating the above fact is that a binary tree can be recovered from the *combination of two total orders*, the one induced by the inorder traversal and the one induced by the postorder traversal. Combining these two total orders yields a partial order, i.e., left descendant, from which the left child of each vertex can be identified. This partial order is shown on the right of Ex. 2.1. See App. B and (Knuth, 1997, §2.3.1, Ex. 7) for further discussion. In light of these observations, we conceive an order-theoretic treatment that constructs

a tree by predicting multiple total orders and intersecting them. In terms of computation, predicting total orders only requires labeling each node with real numbers and then sorting, the complexity of which is linear under radix sort. On the other hand, an arc-factored model necessarily computes all $\mathcal{O}(N^2)$ pair-wise scores for every pair of vertices to decide the existence of each edge.

Next, we generalize the intuitions gained from this example. In §3, we explore the class of graphs that can be efficiently represented with partial orders. In §4, we show how to learn the ordering efficiently with neural networks.

## 3 Order and Structure

In this section, we describe an order-theoretic treatment for linguistic structure prediction. Specifically, we treat the structure to be predicted as a partially ordered set, i.e., a set equipped with a **transitive** relation $\prec$. We begin by revisiting how linguistic structures are represented as graphs.

### 3.1 Linguistic Structures as Directed Graphs

Let $\Sigma$ be an alphabet, i.e., a finite set of natural language tokens, and let $\boldsymbol{w} = w_1 w_2 \cdots w_N \in \Sigma^*$ be a string. Linguistic structure prediction is the task of assigning a structure, e.g., a dependency tree, to a given string $\boldsymbol{w}$ in natural language.

A wide range of linguistic structures are built upon the relations between pairs of tokens. Many structured prediction models are thus arc-factored, i.e., they predict the arcs between a pair of tokens and then combine them back into structures, which are our focus in this work. Formally, their major goal is to model the homogeneous relation[1] on the **spanning node set** $V = \{w_1, w_2, \cdots, w_N\}$ of a sentence $\boldsymbol{w} = w_1 \cdots w_N$ (Kübler et al., 2009). The output space is defined by the input itself, in contrast to the external label spaces in other tasks such as classification or language generation.

**Definition 3.1** (Structure)**.** *A **structure** over a string $\boldsymbol{w} = w_1 w_2 \cdots w_N$ is a directed graph $\mathcal{G} = (V, E)$, where $V = \{w_1, w_2, \cdots, w_N\}$, $E \subseteq V \times V$ is the set of arcs. A **typed structure** $\mathcal{G} = (V, E, R)$ is a structure with $E \subseteq V \times V \times R$, where $R$ is a finite set of relation labels.*

---

[1] A **homogeneous relation** on a set $\mathcal{X}$ is a binary relation between two elements in $\mathcal{X}$. It can be equivalently represented with the set of edges in a graph in which $\mathcal{X}$ is the set of vertices.

Most linguistic structures are naturally subsumed under this definition. We give two examples of linguistic structure prediction tasks.

**Example 3.2** (Dependency Parsing; Kübler et al., 2009, Def. 2.3)**.** A **dependency structure** is a structure $\mathcal{G} = (V, E, R)$, where $E \subseteq V \times V \times R$, and $R$ is the set of dependency relation types. If $(x, y, r) \in E$, then $\forall r' \neq r, (x, y, r') \notin E$. ■

**Example 3.3** (Coreference Resolution)**.** A **coreference structure** is a structure $\mathcal{G} = (V, E, R)$, where $E \subseteq V \times V \times R$, and $R = \{r, r'\}$. The relations $r, r'$ represent the entity mention and coreference, respectively. We have $(x, y, r) \in E$ if and only if the textual span $x : y$ in $\boldsymbol{w}$ is a mention of an entity. $(x_1, x_2, r') \in E \land (y_1, y_2, r') \in E$ if and only if the textual spans $x_1 : y_1$ and $x_2 : y_2$ corefer. ■

### 3.2 From Directed Graphs to Partial Orders

Our treatment constructs linguistic structures with techniques from order theory. The key is to cast the relation between tokens as an order, which is defined as follows.

**Definition 3.4** (Order; Hausdorff, 1914)**.** *An **order** over a set $V$ is a relation $\prec$ such that the following hold for all $x, y, z \in V$:*

 (a) ***irreflexivity***: $x \not\prec x$;

 (b) ***asymmetry***: $x \prec y \implies y \not\prec x$;

 (c) ***transitivity***: $x \prec y \land y \prec z \implies x \prec z$.

Natural language exhibits structural sparsity in that each token in a string usually only interacts with very few other tokens with a particular relation. For instance, in a dependency graph, there are no direct paths between most of the word pairs. Such sparsity, from an order-theoretic point of view, can be characterized by **incomparability** in a partially ordered set (Birkhoff, 1967, Chapter 1, p. 2).

By analogy, we define the following **partially ordered structure**, which is a partially ordered set mathematically. Its elements are the tokens of a string, and its order encodes a linguistic structure.

**Definition 3.5** (Partially Ordered Structure)**.** *Let $\mathcal{G} = (V, E)$ be a structure. Define the following relation $\prec$: For $x, y \in V$, $x \prec y \iff (x, y) \in E$. We call $\mathcal{P} = (V, E, \prec)$ a **partially ordered structure** if $\prec$ satisfies Def. 3.4.*

The essential theoretical foundation of our linguistic structure prediction framework is the classic result that partial orders can be represented by an intersection of total orders (Dushnik and Miller, 1941). It is this result that enables us to use real numbers as a proxy to determine the partial ordering of tokens.

**Definition 3.6** (Totally Ordered Structure)**.** *A partially ordered structure $\mathcal{P} = (V, E, \prec)$ is **totally ordered** if $\forall x, y \in V : x \prec y \lor y \prec x$.*

Due to the transitivity of the ordering relation $\prec$, a totally ordered structure of $|V|$ elements always contains $|E| = \binom{|V|}{2}$ relations. Given a collection of structures $\{(V, E_k)\}_{k \in [K]}$ defined over the same set of vertices $V$, their **intersection** is also a structure—namely $(V, \cap_{k \in [K]} E_k)$, where $K \in \mathbb{N}, [K] \stackrel{\text{def}}{=} \{1, \cdots, K\}$. The intersection of partially ordered structures remains partially ordered.

We now cite a famous theorem from order theory.

**Theorem 3.7** (Szpilrajn (1930))**.** *Every partially ordered structure is contained in a totally ordered structure, i.e., for every partially ordered structure $\mathcal{P} = (V, E, \prec)$, there exists a totally ordered structure $\mathcal{T} = (V, \widehat{E}, \prec)$ such that $E \subseteq \widehat{E}$.*

Thm. 3.7 ensures that every partially ordered structure can be embedded in some totally ordered structure in the sense that the totally ordered structure *contains* all the relations in the partially ordered structure. More importantly, a stronger result can be shown: Partially ordered structures can *always* be represented as intersections of a collection of totally ordered structures.

**Definition 3.8** (Realizer)**.** *Let $\mathcal{P} = (V, E, \prec)$ be a partially ordered structure. A **realizer** $R_{\mathcal{P}}$ of $\mathcal{P}$ is a set of totally ordered structures $\{\mathcal{T}_1, \mathcal{T}_2, \cdots, \mathcal{T}_K\}$ over $V$, i.e., each $\mathcal{T}_k = (V, E_k, \prec_k)$, such that $E = \bigcap_{k \in [K]} E_k$. In other words, $\forall x, y \in V, x \prec y \iff \bigwedge_{k \in [K]} x \prec_k y$.*

**Theorem 3.9** (Dushnik and Miller, 1941, Thm. 2.32)**.** *There exists a realizer $R_{\mathcal{P}}$ for every partially ordered structure $\mathcal{P} = (V, E, \prec)$.*

A corollary of the above theorem is that the complexity of a partially ordered structure can be characterized by its order dimension, which is defined as follows.

**Definition 3.10** (Order Dimension; Dushnik and Miller, 1941)**.** *Let $\mathcal{P} = (V, E, \prec)$ be a partially ordered structure. The **order dimension** $D_{\mathcal{P}}$ of $\mathcal{P}$ is the cardinality of the smallest realizer of $\mathcal{P}$.*

### 3.3 Efficiency Guarantees

In this section, we give an efficiency guarantee of order-theoretic structured prediction. These efficiency guarantees come from a series of results in

order theory and lattice theory (Dushnik and Miller, 1941; Hiraguchi, 1955; Birkhoff, 1967, *inter alia*).

First, it is important to note that *not* all partially ordered structures can be represented as an intersection of a *constant* number of totally ordered structures (Dushnik and Miller, 1941, Thm. 4.1).

In fact, testing whether the order dimension of a partial order $\mathcal{P}$ is at most $K$, $\forall K \geq 3$ is NP-complete (Yannakakis, 1982). However, we contend that most of the linguistic structures found in natural language processing (Smith, 2011)—including trees, equivalence classes (i.e., set partitioning), and alignment (i.e., bipartite matching)—can be represented as the intersection of 2 totally ordered structures. We postulate that this is possible due to their innate sparsity, i.e., a token tends to only interact with a few other tokens. These assumptions are formalized as follows.

**Assumption 3.11** (Sparsity). *A class of linguistic structures $\mathcal{G} = (V, E)$ over natural language strings $\boldsymbol{w} \in \Sigma^*$ with $N = |\boldsymbol{w}|$ is called **sparse** if $\mathcal{O}(|E|) = \mathcal{O}(N)$.*

**Assumption 3.12** (Linguistic Structures are 2-dimensional). *Structures in natural language can be represented as intersections of 2 totally ordered structures.*

We justify Assumptions 3.11–3.12 in App. D. Empirical evidence is also provided in §5, where 2-dimensional order-theoretic models are trained to tackle two linguistic structure prediction tasks with high performance.

### 3.4 Token-Split Structures

An obvious limitation of our formulation of linguistic structures as partial orders is that by Def. 3.4, partial order is transitive. In other words, $x \prec y \wedge y \prec z$ implies $x \prec z$, which, however, does *not* hold in the structures characterized by the directed graph formalization in Def. 3.1. In addition, we note that our notation of structures generalizes to cyclic graphs. However, partially ordered structures are inherently acyclic due to the transitivity of $\prec$. We now introduce the **token-split structure**, which enables cycles and removes redundant edges introduced by transitivity in partially ordered structures.

**Definition 3.13** (Token-Split Structure). *A **token-split structure** induced by a structure $\mathcal{G} = (V, E)$ is a structure $\mathcal{P} = (\widehat{V}, \widehat{E}, \prec)$ such that*

(a) $\widehat{V} \stackrel{\text{def}}{=} V^r \cup V^b$, *where $V^r = \{x^r \mid x \in V\}, V^b = \{x^b \mid x \in V\}$;*

(b) $V^r \cap V^b = \varnothing$;

(c) $\widehat{E} = \left\{ (x^r, y^b) \mid (x, y) \in E \right\}$.

In other words, a token-split structure maps the edges from the original structure, *including self-loops*, into a bipartite graph in which the edges are oriented from $V^r$ to $V^b$. An example is displayed in Fig. 1b.

Given a token-split structure $\mathcal{P} = (\widehat{V}, \widehat{E}, \prec)$, we can recover the original structure $\mathcal{G} = (V, E)$ from which $\mathcal{P}$ is induced using the following equation

$$E = \{(x, y) \mid x^r \in V^r \wedge y^b \in V^b \wedge x^r \prec y^b\} \quad (1)$$

**Theorem 3.14.** *Token-split structures are partially ordered.*

*Proof.* See App. C.1. ∎

**Remark 3.15** (Conversion between Structures and Partially Ordered Structures). *Thm. 3.14 and Eq. (1) ensure that we can convert back and forth between any structure under Def. 3.1 and a partially ordered structure. Specifically, they enable us to first convert a structure to a partially ordered structure, predict it order-theoretically, and then finally convert it back to a structure.*

## 4 A Neural Parameterization

In this section, we describe the core technical contribution of our work. We show how to model partially ordered structures with a neural model. Specifically, we define a parameterized realizer of Def. 3.8 and an objective function for training the realizer to model the token-split structures. We also give algorithms for efficient training and decoding.

### 4.1 Neuralized Total Order

We now discuss a parameterized neural network that induces partial orders as the intersection of several total orders.

**Definition 4.1** (Functional Realizer). *A **functional realizer** of a partially ordered structure $\mathcal{P} = (V, E, \prec)$ is a set of mappings $\mathcal{F}_{\boldsymbol{\theta}} = \{f_{\boldsymbol{\theta}}^{(1)}, \cdots, f_{\boldsymbol{\theta}}^{(K)}\}$, where $\boldsymbol{\theta}$ is the set of learnable parameters shared among $f_{\boldsymbol{\theta}}^{(k)}$, and the order dimension $K \in \mathbb{N}$ is a hyperparameter of the realizer. The realize element $f_{\boldsymbol{\theta}}^{(k)} \colon V \to \mathbb{R}$, $\forall k \in [K]$ maps each vertex in the input structure to a real number. We overload $\mathcal{F}_{\boldsymbol{\theta}}$ as a mapping $\mathcal{F}_{\boldsymbol{\theta}} \colon V \to \mathbb{R}^K$, defined as $\mathcal{F}_{\boldsymbol{\theta}}(x) \stackrel{\text{def}}{=} \left[ f_{\boldsymbol{\theta}}^{(1)}(x), \cdots, f_{\boldsymbol{\theta}}^{(K)}(x) \right]^{\top}$.*

The set of real numbers $\mathbb{R}$ is totally ordered, in which the order is given by the $<$ (less than) relation. Each individual $f_{\boldsymbol{\theta}}^{(k)} \in \mathcal{F}_{\boldsymbol{\theta}}$ induces a total order $\mathcal{T}_k = \left(V, \{(x, y) \mid x, y \in V, f_{\boldsymbol{\theta}}^{(k)}(x) < f_{\boldsymbol{\theta}}^{(k)}(y)\}, \prec_k\right)$.[2]

The functional realizer assigns $K$ total orders $\{\mathcal{T}_1, \mathcal{T}_2, \cdots, \mathcal{T}_K\}$ to the input string. During decoding, an edge from $x$ to $y$ exists in $\mathcal{P}$ if and only if $x \prec_k y$ holds in $\mathcal{T}_k, \forall k \in [K]$.

Implementing Def. 4.1 with neural networks is straightforward. To obtain $f_{\boldsymbol{\theta}}^{(k)}(x^r)$ and $f_{\boldsymbol{\theta}}^{(k)}(x^b)$, where $x^r, x^b$ are two vertices introduced by the token-split formulation (Def. 3.13) corresponding to the same token $\boldsymbol{w}_x$ in the input, we apply two linear projections on the contextualized representation of $x$ given by a pretrained model parameterized by $\boldsymbol{\theta}$.[3] In total, $2K$ real numbers are predicted for each input token.

## 4.2 Learning a Functional Realizer

To learn the functional realizers with a gradient-based procedure, we need a differentiable objective. In a partially ordered structure $\mathcal{P}$ with functional realizer $\mathcal{F}_{\boldsymbol{\theta}} = \{f_{\boldsymbol{\theta}}^{(1)}, f_{\boldsymbol{\theta}}^{(2)}, \cdots, f_{\boldsymbol{\theta}}^{(K)}\}$, we have $x \prec y$ if and only if $\bigwedge_{k \in [K]} \left(f_{\boldsymbol{\theta}}^{(k)}(x) < f_{\boldsymbol{\theta}}^{(k)}(y)\right)$. We re-express this condition as follows:

$$F_{\boldsymbol{\theta}}(x, y) \stackrel{\text{def}}{=} \max_{k \in [K]} \left(f_{\boldsymbol{\theta}}^{(k)}(x) - f_{\boldsymbol{\theta}}^{(k)}(y)\right) < 0 \quad (2)$$

We call $F_{\boldsymbol{\theta}}$ a **pair-wise function**. On the other hand, we have $x \not\prec y$ if and only if $\bigvee_{k \in [K]} \left(f_{\boldsymbol{\theta}}^{(k)}(x) \geq f_{\boldsymbol{\theta}}^{(k)}(y)\right)$. This condition can be re-expressed as $F_{\boldsymbol{\theta}}(x, y) \geq 0$. Thus, empirically, the smaller $F_{\boldsymbol{\theta}}(x, y)$ is, the more likely the relation $x \prec y$ exists.

We now define a training objective, which encourages the model to make decisions that comply with the order constraints enforced by the structures, described by Eq. (2). Given the token-split version $\mathcal{P} = (V, E, \prec)$ induced by the structure being modeled, we consider the following objective

$$\mathcal{L}(\boldsymbol{\theta}) = \log \sum_{(x,y) \in V^2 \setminus E} \exp -F_{\boldsymbol{\theta}}(x, y) + \\ \log \sum_{(x,y) \in E} \exp F_{\boldsymbol{\theta}}(x, y) \quad (3)$$

---

[2]In this work, we assume $f_{\boldsymbol{\theta}}^{(k)}$ is injective, i.e., $\forall x, y \in V, f_{\boldsymbol{\theta}}^{(k)}(x) \neq f_{\boldsymbol{\theta}}^{(k)}(y)$. See §8.4 for further discussion on the practicality of this assumption.

[3]If $\boldsymbol{w}_x$ consists of more than one subword due to tokenization, we apply the projection to the representation of the last subword.

The first term maximizes $F_{\boldsymbol{\theta}}(x, y)$ for $x \not\prec y$, while the second minimizes $F_{\boldsymbol{\theta}}(x, y)$ for $x \prec y$. Note that in the second term, we assume $\mathcal{O}(|E|) = \mathcal{O}(N)$ in a linguistic structure following Assumption 3.11.

## 4.3 An Efficient Algorithm

We remark that both training and decoding of the proposed model can be regarded as performing an aggregation for every token $x \in V$.

**Definition 4.2** (Aggregation). *An $\oplus$-aggregation given a token $x$ for a pair-wise function $F_{\boldsymbol{\theta}}$ over the set $V$ is an operation $\bigoplus_{y \in V} F_{\boldsymbol{\theta}}(x, y)$, where $\oplus$ is a commutative and associative operation over which real number addition $+$ is distributive.*

Aggregation is a common abstraction for computing the relation between a token $x$ and every other token. The aggregation operator is associative and commutative, thus can be computed in parallel. The number of required computations is $\mathcal{O}(|V|)$ for naïvely computing an aggregation of $x$.

During training, we $\oplus$-aggregate using negative `log-sum-exp`, i.e., we compute $-\log \sum_y \exp(-F_{\boldsymbol{\theta}}(x, y))$ for all $x$, to compute the first term of Eq. (3). In greedy decoding, we $\oplus$-aggregate by computing $\min_y F_{\boldsymbol{\theta}}(x, y)$ to find the optimal relation arc from each $x$. Naïvely, $\oplus$-aggregating for every token $x \in V$ takes $\mathcal{O}(N^2)$ in total, as each aggregand has a complexity of $\mathcal{O}(N)$. However, the partial order we assigned over $V$ allows us to efficiently compute the aggregands.

For $K = 2$, we can inspect Eq. (2) to see that $F_{\boldsymbol{\theta}}(x, y)$ is equal to *either* $f_{\boldsymbol{\theta}}^{(1)}(x) - f_{\boldsymbol{\theta}}^{(1)}(y)$ or $f_{\boldsymbol{\theta}}^{(2)}(x) - f_{\boldsymbol{\theta}}^{(2)}(y)$. We now define the following two subsets of $V$ for $k \in \{1, 2\}$:

$$\mathcal{S}_k(x) = \left\{y \mid F_{\boldsymbol{\theta}}(x, y) = f_{\boldsymbol{\theta}}^{(k)}(x) - f_{\boldsymbol{\theta}}^{(k)}(y)\right\}$$

Using this notation, we can write the following

$$\bigoplus_{(x,y) \in V^2} F_{\boldsymbol{\theta}}(x, y) = \bigoplus_{x \in V} \bigoplus_{y \in V} F_{\boldsymbol{\theta}}(x, y) \quad (5a)$$

$$= \underbrace{\bigoplus_{x \in V} \bigoplus_{y \in \mathcal{S}_1(x)} \left(f_{\boldsymbol{\theta}}^{(1)}(x) - f_{\boldsymbol{\theta}}^{(1)}(y)\right)}_{\stackrel{\text{def}}{=} G_1} \quad (5b)$$

$$\oplus \underbrace{\bigoplus_{x \in V} \bigoplus_{y \in \mathcal{S}_2(x)} \left(f_{\boldsymbol{\theta}}^{(2)}(x) - f_{\boldsymbol{\theta}}^{(2)}(y)\right)}_{\stackrel{\text{def}}{=} G_2}$$

We now give an efficient algorithm to compute $G_1$ and, by symmetry, $G_2$. Our first observation is that,

by distributivity, we can write

$$G_1 = \bigoplus_{x \in V} \bigoplus_{y \in \mathcal{S}_1(x)} \left( f_{\boldsymbol{\theta}}^{(1)}(x) - f_{\boldsymbol{\theta}}^{(1)}(y) \right) \quad \text{(6a)}$$

$$= \bigoplus_{x \in V} \underbrace{\left( f_{\boldsymbol{\theta}}^{(1)}(x) + \bigoplus_{y \in \mathcal{S}_1(x)} -f_{\boldsymbol{\theta}}^{(1)}(y) \right)}_{\stackrel{\text{def}}{=} G_1(x)} \quad \text{(6b)}$$

Alone, this application of dynamic programming does not reduce the complexity from $\mathcal{O}(N^2)$ to $\mathcal{O}(N)$ as desired because the inner aggregand, $\bigoplus_{y \in \mathcal{S}_1(x)} -f_{\boldsymbol{\theta}}^{(1)}(y)$, itself still takes $\mathcal{O}(N)$ time. However, we are able to compute $\bigoplus_{y \in \mathcal{S}_1(x)} -f_{\boldsymbol{\theta}}^{(1)}(y)$ in amortized $\mathcal{O}(1)$ time due to Fredman's (1976, Eq. 1) algebraic trick.

The strategy is to sort[4] the vertices of the partially ordered structure according to $f_{\boldsymbol{\theta}}^{(1)}(y) - f_{\boldsymbol{\theta}}^{(2)}(y)$. Thus, if we have $f_{\boldsymbol{\theta}}^{(1)}(y) - f_{\boldsymbol{\theta}}^{(2)}(y) < f_{\boldsymbol{\theta}}^{(1)}(x) - f_{\boldsymbol{\theta}}^{(2)}(x)$, simple algebra reveals that $f_{\boldsymbol{\theta}}^{(2)}(x) - f_{\boldsymbol{\theta}}^{(2)}(y) < f_{\boldsymbol{\theta}}^{(1)}(x) - f_{\boldsymbol{\theta}}^{(1)}(y)$. Thus, for a given $x$, every vertex $y$ that comes *before* $x$ in the sorted order satisfies $F_{\boldsymbol{\theta}}(x, y) = f_{\boldsymbol{\theta}}^{(1)}(x) - f_{\boldsymbol{\theta}}^{(1)}(y)$. Aggregating in this order enables intermediate results to be *reused*.

---

**Algorithm 1** Computing $G_1$ when $K = 2$.

---

1: **procedure** COMPUTE-$G_1(f_{\boldsymbol{\theta}}^{(1)}, f_{\boldsymbol{\theta}}^{(2)}, V)$
2: $\quad U \leftarrow \texttt{sort}\left(V, \texttt{key} = f_{\boldsymbol{\theta}}^{(1)} - f_{\boldsymbol{\theta}}^{(2)}\right)$
3: $\quad G_1, s_1 \leftarrow \mathbf{0}, \mathbf{0} \triangleright \mathbf{0}$ *is the zero element of* $\oplus$
4: $\quad$ **for** $n = 1$ **up to** $N$ :
5: $\quad\quad q_1 = f_{\boldsymbol{\theta}}^{(1)}(U_n) + s_1 \triangleright q_1 = G_1(U_n)$
6: $\quad\quad G_1 \oplus= q_1$
7: $\quad\quad s_1 \oplus= -f_{\boldsymbol{\theta}}^{(1)}(U_n)$
8: $\quad$ **return** $G_1$

---

Likewise, if we sorted in reverse, i.e., according to $f_{\boldsymbol{\theta}}^{(2)}(y) - f_{\boldsymbol{\theta}}^{(1)}(y)$, the same manipulation yields that for a given $x$, every vertex $y$ that comes *before* $x$ in the *reverse* sorted order satisfies $F_{\boldsymbol{\theta}}(x, y) = f_{\boldsymbol{\theta}}^{(2)}(x) - f_{\boldsymbol{\theta}}^{(2)}(y)$.

The algorithm for computing $G_1$ is given in Algorithm 1, which has $\mathcal{O}(N)$ computations in total. Moreover, if parallelized, it can be run in $\mathcal{O}(\log N)$ time. For $K > 2$, we speculate that the aggregation algorithm can be done in $\mathcal{O}(KN \log^{K-2} N)$. We leave this to future work. See App. E.2 for further discussion.

---

[4] As before, we take the complexity of sorting to be $\mathcal{O}(N)$ where we can apply radix sort as implemented by Pytorch.

# 5 Experiments

We report the experimental results on two representative linguistic structure prediction problems in NLP, namely dependency parsing and coreference resolution. The graph-theoretic definitions of these tasks are given in Examples 3.2 and 3.3. We first convert the linguistic structures to partially ordered (token-split) structures as described in §3.4, and then apply the neural method described in §4 to model the partially ordered structures.

## 5.1 Dependency Parsing

**Modeling.** Orders $\prec$ are not typed in Def. 3.5. In other words, under Def. 3.5, all relations in a partially ordered structure are of the same type. To model dependency type labels, we apply a token-level classifier on the contextualized representation. During decoding, similar to arc-factored models for dependency parsing, we keep the head word that minimizes $F_{\boldsymbol{\theta}}(x, y)$ for a given $x$, i.e., $\arg\min_{y \in V} F_{\boldsymbol{\theta}}(x, y)$.

For pretrained language models, we use XLNet-large-cased[5] (Yang et al., 2019) for PTB, bert-base-chinese[6] for CTB, and bert-base-multilingual-cased[7] for UD.

**Datasets.** We conduct experiments on the English Penn Treebank (PTB; Marcus et al., 1993), the Chinese Penn Treebank (CTB; Xue et al., 2005), and the Universal Dependencies 2.2 (UD; Nivre et al., 2018). Hyperparameter settings and dataset statistics are given in Apps. F.1 and G.1.

**Accuracy.** We report the experimental results in Tab. 1. The full results on UD are included in App. I.1. On PTB and UD, our method achieves state-of-the-art performance compared with $\mathcal{O}(N^3)$ (Yang and Tu, 2022), $\mathcal{O}(N^2)$ (Mrini et al., 2020), and $\mathcal{O}(N)$ (Amini et al., 2023) methods. Although Amini et al.'s (2023) method has the same complexity as ours, it is worth noting that our method is more general since it can handle non-projective dependencies *without* using pseudo-projectivization (Nivre and Nilsson, 2005).

**Efficiency.** We evaluate the efficiency of our method with two representative baseline models. As depicted in Tab. 2, we observe that our method with $K = 2$ is almost 10 times as fast as Biaff

---

[5] https://huggingface.co/xlnet-large-cased
[6] https://huggingface.co/bert-base-chinese
[7] https://huggingface.co/bert-base-multilingual-cased

| Model | PTB | | CTB | | UD |
|---|---|---|---|---|---|
| | UAS | LAS | UAS | LAS | LAS |
| Zhou and Zhao* | 97.0 | 95.4 | 91.2 | 89.2 | - |
| Mrini et al.* | **97.4** | 96.3 | 94.6 | 89.3 | - |
| Chen and Manning | 91.8 | 89.6 | 83.9 | 82.4 | - |
| Dozat and Manning | 95.7 | 94.1 | 89.3 | 88.2 | 91.8 |
| Yang and Tu# | **97.4** | 95.8 | **93.3** | **92.3** | 91.9 |
| Amini et al. | **97.4** | 96.4 | 93.2 | 91.9 | 91.8 |
| Ours ($K = 2$) | 97.1 | 96.1 | 90.7 | 89.5 | 91.2 |
| Ours ($K = 4$) | **97.4** | **96.4** | 92.4 | 91.4 | **92.1** |

Table 1: Experimental results on PTB, CTB, and UD. * indicates usage of extra constituency annotation. # is our re-implementation using the same pretrained encoder as ours. $K$ is the dimension of the realizer used.

| #token | Speed (sent/s) ↑ | | | Memory (GB) ↓ | | |
|---|---|---|---|---|---|---|
| | Ours | Hexa | Biaff | Ours | Hexa | Biaff |
| 32 | 3232 | 2916 | 493 | 1.7 | 2.9 | 4.5 |
| 64 | 3332 | 3011 | 328 | 1.7 | 3.0 | 10.1 |
| 128 | 3182 | 2649 | 202 | 1.9 | 3.7 | 30.6 |
| 256 | 3314 | 3270 | 98 | 3.1 | 4.5 | 56.2 |
| overall | **3347** | 3176 | 338 | **1.7** | 3.0 | 10.6 |

Table 2: Speed and memory consumption comparison on PTB test set. We report results averaged over 3 random runs of our method with $K = 2$. The other settings and the results for Hexa and Biaff are taken from Amini et al. (2023, Tab. 3).

(Dozat and Manning, 2017), and consumes less memory than Hexa (Amini et al., 2023), which is $\mathcal{O}(N)$ in space complexity. We further include some qualitative examples using $K = 2$ in App. J for a more intuitive picture of our method.

## 5.2 Coreference Resolution

**Modeling.** Our method operates in a two-stage manner to accommodate the two relations in Ex. 3.3. First, it extracts a list of entity mentions using the partial order induced by $r$ (mention relation). In other words, $x \prec y \iff$ span $x : y$ is an entity mention. Then, it models the partial order induced by $r'$ (coreference relation) over the extracted mentions. In other words, $m_1 \prec m_2 \iff$ mention $m_1$ corefers to $m_2$. To find the optimal coreferent antecedent for each mention $m$, we keep $m'$ that minimizes $F_{\boldsymbol{\theta}}(m, m')$.

The overall complexity of the coreference resolution model is $\mathcal{O}(N)$, since the complexity of the encoder used (Beltagy et al., 2020) and the number of valid mentions are both $\mathcal{O}(N)$,

assuming entity mentions are constituents (Liu et al., 2022). We experiment on the CoNLL-2012 English shared task dataset (OntoNotes; Pradhan et al., 2012). Hyperparameter settings and dataset statistics are given in Apps. F.2 and G.2.

**Accuracy.** The experimental results are displayed in Tab. 3. Similar to the results for dependency parsing, an intersection of 2 total orders can already achieve reasonable performance on coreference resolution. This provides *empirical evidence* for our assertion in §3.3 that most structures in NLP can be represented as the intersection of at most 2 total orders. When $K = 4$, the performance of our method is comparable to Kirstain et al. (2021), which uses the same pretrained encoder as ours and requires an $\mathcal{O}(N^2)$ biaffine product computation for token-pair scores.

**Efficiency.** We compare the efficiency of our method with Kirstain et al.'s (2021) method. It is worth noting that Kirstain et al. (2021) has already performed aggressive optimization in both the speed and memory footprint of coreference modeling. Our method is still 2 times as fast, achieving a speed of 82.8 documents per second vs. 41.9, while using less memory, especially on long documents. The full efficiency statistics are given in App. H.

| | Avg. P | Avg. R | Avg. F1 |
|---|---|---|---|
| Lee et al. (2017) | 69.9 | 64.7 | 67.2 |
| Kantor and Globerson | 76.1 | 77.1 | 76.6 |
| Joshi et al. (2020) | 80.1 | 78.9 | 79.6 |
| Xu and Choi (2020) | 80.3 | 79.5 | 79.9 |
| Kirstain et al. (2021) | 81.2 | 79.4 | 80.3 |
| Ours ($K = 2$) | 75.2 | 74.8 | 75.0 |
| Ours ($K = 4$) | 79.3 | 79.0 | 79.2 |

Table 3: Experimental results on the OntoNotes benchmark. $K$ is the dimension of the realizer.

## 6 Related Work[8]

### 6.1 Structured Prediction

Structured prediction constitutes an important part of natural language processing. It involves the modeling of interrelated variables or outputs with structural constraints. Some representative structured prediction problems are sequence tagging (Church, 1988), dependency parsing (Kübler et al., 2009), and coreference resolution (Stede, 2012).

---

[8]More related work is included in App. A.

Structured prediction can often be formulated as learning and inference of probabilistic graphical models (Smith, 2011, §2.2). The key idea is to represent the probability distribution over the output space using a graph, in which each vertex corresponds to a random variable, and each edge corresponds to a dependence relation between two random variables.

## 6.2 Graph-Based Parsing

Graph-based parsers, or arc-factored parsers, construct graphs by scoring all possible arcs (Eisner, 1996; McDonald and Pereira, 2006) between each pair of words. At inference time, they use either maximum spanning tree (MST) finding algorithms (Chu and Liu, 1965; Edmonds, 1967; Tarjan, 1977), or the projective MST algorithm (Eisner, 1996) to build the valid dependency trees with maximum score. Kiperwasser and Goldberg (2016) present a neural graph-based parser that uses the same kind of attention mechanism as Bahdanau et al. (2015) for computing arc scores. Greedy decoding that independently assigns a head word to each word (Dozat and Manning, 2017) is also widely used as an approximation to exact inference algorithms.

## 6.3 Tagging-Based Parsing

Inspired by transition-based parsers (Knuth, 1965) and Bangalore and Joshi's (1999) seminal work on **supertagging**, one line of work uses pretrained models to parse dependency trees by inferring tags for each word in the input sequence. Li et al. (2018) and Kiperwasser and Ballesteros (2018) predict the relative position of the dependent with respect to its head in a sequence-to-sequence manner. Strzyz et al. (2019) give a framework for analyzing similar tagging schemes. Gómez-Rodríguez et al. (2020) infer a chunk of actions in a transition-based system for each word in the sequence.

For non-projective dependency parsing, Gómez-Rodríguez and Nivre (2010, 2013) show that efficient parsers exist for 2-planar trees (Yli-Jyrä, 2003), a sub-class of non-projective trees whose arcs can be partitioned into 2 sets while arcs in the same set do not cross each other. Strzyz et al. (2020) propose an encoding scheme for 2-planar trees, enabling a tagging-based parser for such trees. As mentioned in §2.1, to handle the entire set of non-projective trees, the size of the tag set has to be unrestricted, which limits the efficiency and applicability of this series of approaches of approaches.

## 6.4 Parsing with Syntactic Distance

Shen et al. (2018a,b) introduce a constituent parsing scheme which is also based on the comparison of real numbers. In this scheme, a neural model is trained to assign one real number, termed the **syntactic distance**, to the gap between every pair of neighboring tokens. To parse a span into two sub-constituents, the gap with the largest syntactic distance within that span is chosen as the split point. Parsing can be done by recursively performing the above splitting procedure starting from a given string. The algorithm has a runtime complexity of $\mathcal{O}(N \log N)$, which is significantly more efficient than chart-based parsers with $\mathcal{O}(N^2)$ complexity. However, this method does not generalize easily to perform non-context-free parsing, since it cannot handle the possible discontinuity of constituents. Moreover, the recursive splitting procedure restricts the output space of parse trees to be a subset of phrase-structure trees (Dyer et al., 2019).

## 7 Conclusion

In this paper, we propose an order-theoretic treatment of linguistic structured prediction. Theoretical and empirical results show that most linguistic structure prediction problems can be solved in linear time and memory by framing them as partial orderings of the tokens in the input string. We demonstrate the effectiveness of our method on dependency parsing and coreference resolution, setting the new state-of-the-art accuracy in some cases and achieving significant efficiency improvements.

## 8 Limitations

### 8.1 Decoding Algorithms

This work does not provide algorithms for particular structures or algorithms that ensure the well-formedness of structures, such as maximum spanning trees and projective trees. It remains to be investigated whether existing constrained decoding algorithms for arc-factored models (Chu and Liu, 1965; Edmonds, 1967; Eisner, 1997, *inter alia*) have their counterparts in the order-theoretic method. We would like to explore decoding algorithms for structured prediction under order-theoretic formulation in future work.

### 8.2 Interpretability

In our method, the interactions between tokens are not directly modeled as in graph-based structured

prediction models, which makes it more difficult to interpret the output of our model. In addition, we leave to future work the investigation of the total ordering metrics (see App. J) learned by the realizers in our method.

### 8.3 Hardness of Learning

Intuitively, it is harder to learn partial orders over strings than directly modeling the arcs in a graph, since our order-theoretic treatment has much fewer parameters when scoring token pairs. We also observed in our experiments that order-theoretic models take more training iterations to converge than arc-factored models.

For instance, consider the modeling of a tree structure with $N$ nodes with $N-1$ arcs using partial order, which implies $N-1$ constraints of the form $x \prec y$ and $N^2 - 2N + 1$ constraints of $x \not\prec y$. From a theoretical perspective, $K = 2$ is sufficient to represent such a structure as shown in §3. In other words, there always exist 2 total orders whose intersection satisfies the aforementioned $N(N-1)$ constraints. However, it might not be easy to find such orders in practice.

A realizer with $K$ beyond 2 can more easily satisfy the constraints, especially of the form $x \not\prec y$— since there are more constraints of this form. It allows more possibilities for $\bigvee_{k \in [K]} f_{\boldsymbol{\theta}}^{(k)}(x) \geq f_{\boldsymbol{\theta}}^{(k)}(y)$ (i.e., more choices of $k$ to satisfy the expression). On the other hand, using a small $K$ might make it harder to satisfy the constraints.

We plan to further investigate the hardness of learning a string partial order in future work.

### 8.4 Precision of floating-point numbers and numerical stability

Our method might be affected by the finite precision of floating-point numbers and numerical instability when applied to very long strings. Although we did not encounter such issues in our experiments ($N \leq 4096 = 2^{12}$), issues might arise when $N > 65536 = 2^{16}$ if bfloat16 or half precision is used. In such extreme cases, our assumption that $\forall k \in [K], f_{\boldsymbol{\theta}}^{(k)}$ is injective cannot be fulfilled. Thus, *not all* totally ordered structures of $N$ elements can be represented, and our method might not exhibit the desired behavior.

### Ethics Statement

We do not believe the work presented here further amplifies biases already present in the datasets and pretrained models. Therefore, we foresee no ethical concerns in this work.

### Acknowledgments

We would like to thank Zhaofeng Wu, Clément Guerner, and Tim Vieira for their invaluable feedback. We are grateful to the anonymous reviewers for their insightful comments and suggestions. Afra Amini is supported by ETH AI Center doctoral fellowship. MS acknowledges support from the Swiss National Science Foundation (Project No. 197155), a Responsible AI grant by the Haslerstiftung; and an ETH Grant (ETH-19 21-1).

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

## A    Related Work

### A.1    Ordinal Regression

Ordinal regression is a family of problems that involve ranking a set of objects. Unlike classification, the label spaces in ordinal regression exhibit some natural ordering in its elements (McCullagh, 1980). For instance, in information retrieval, a ranking model sorts a set of documents typically according to the document's relevance to the query. Practically, ordinal regression can either be tackled as either regression or classification by treating the ranks as real-values or the assignment to a particular rank value as a classification (Shawe-Taylor and Cristianini, 2004).

### A.2    Order Embeddings of Lexicons

The notion of partial order has also been explored for learning word embeddings. The lexicons of natural languages exhibit hierarchical structures according to the concepts that the words represent (Miller, 1994). For instance, 'cat' and 'dog' are 'animal', 'animal' and 'plant' are 'living thing'. Order embeddings (Vendrov et al., 2015; Athiwaratkun and Wilson, 2018) propose to learn such property by learning embeddings that encode such partial order on the lexicon, resulting in improved performance on downstream tasks such as image caption retrieval.

## B    An Order-Theoretic Re-evaluation of §2.2

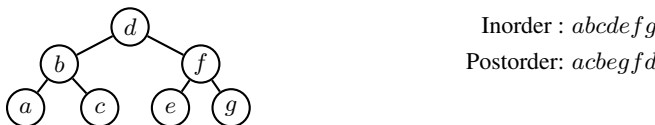

Inorder : $abcdefg$
Postorder: $acbegfd$

(a) The example binary tree in §2.2 and its traversal sequences.

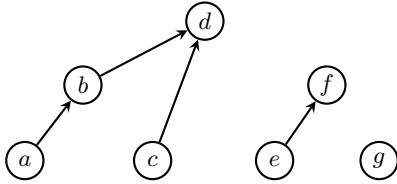

(b) Partial order of §2.2 defined by the intersection of in- and postorder. $A \to B$ represents the relation $A \prec_1 B$.

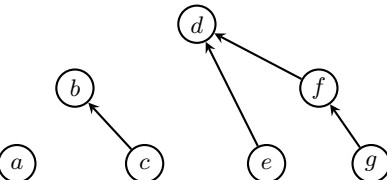

(c) Partial order of §2.2 defined by the intersection of reversed in- and postorder. $A \to B$ represents the relation $A \prec_2 B$.

Figure 3: An order-theoretic re-evaluation of Thm. B.1.

**Theorem B.1** (A binary tree and its traversal; Knuth, 1997, §2.3.1, Ex. 7)**.** *Given the inorder and either the pre- or postorder traversal of the vertices in a binary tree, the binary tree structure can be reconstructed.*

*Proof Sketch (order-theoretic).*  Without loss of generality, we explain the case of the combination of in- and postorder. $V$ denotes the set of vertices in the binary tree. First, the intersection of in- and postorder defines a partial order relation $\mathcal{P}_1 = (V, E_1, \prec_1)$. For any 2 vertices $x, y$ in the binary tree, $x \prec_1 y$ if and only if $x$ is a left descendant of $y$. I.e., $x$ is either the left child or a descendant of the left child of $y$ (see Fig. 3b). Since $x$ is visited before visiting $y$ in both inorder traversal and postorder traversal, if and only if $x$ is the *left* descendant of $y$. The left child of each vertex in $V$ can be decoded from $\mathcal{P}_1$ by finding

the child with the deepest subtree. Second, the intersection of *reversed* inorder and postorder defines a partial order relation $\mathcal{P}_2 = (V, E_2, \prec_2)$. For any 2 vertices $x, y$ in the binary tree, $x \prec_2 y$ if and only if $x$ is a right descendant of $y$ (see Fig. 3c). Since $x$ is visited before visiting $y$ in both the reversed inorder traversal and postorder traversal, if and only if $x$ is the *right* descendant of $y$. The right child of each vertex in $V$ can be decoded from $\mathcal{P}_2$ also by finding the child with the deepest subtree. Thus, the original binary tree can be reconstructed.

∎

## C   Proofs on the Partially Ordered Properties of Structures

### C.1   Proof of Thm. 3.14

**Theorem 3.14.** *Token-split structures are partially ordered.*

*Proof.* We show that a token-split structure $\mathcal{P} = \left(\widehat{V}, \widehat{E}, \prec\right)$ satisfies all the properties of partially ordered structure defined in Def. 3.4.

1. irreflexivity: By Def. 3.13 (c), for all $x \in \widehat{V}$, $x \nprec x$.
2. asymmetry: Suppose that $\exists x, y, x \neq y$, s.t. $x \prec y \wedge y \prec x$. By Definitions 3.13 (b) and 3.13 (c), $x, y \in V^r \cap V^b = \varnothing$. Thus, $x \prec y \implies y \nprec x$.
3. transitivity: $x \prec y \wedge y \prec z$ cannot hold by Def. 3.13 (c). Since $x \prec y$ implies $x \in V^r \wedge y \in V^b$, while $y \prec z$ implies $y \in V^r \wedge x \in V^b$, a contradiction occurs due to $y \in V^r \cap V^b = \varnothing$ by Def. 3.13 (b). $x \prec y \wedge y \prec z \implies x \prec z$ holds since the antecedent of the proposition is always false.

Thus, token-split structures are partially ordered.

∎

## D   Guarantees of Order Dimension of Linguistic Structures

We justify the guarantees of order dimension of linguistic structures. One conventional way to characterize the dimension of partial orders is from a lattice-theoretic point of view. A basic result tells us that a partial order is 2-dimensional *if and only if* its **complete lattice embedding** has a **planar Hasse diagram** (Baker et al., 1972). In other words, its complete lattice embedding can be drawn on a plane without any crossing edges.

**Theorem D.1** (Baker et al., 1972, Thm. 4.1)**.** *Suppose $\mathcal{P} = (V, E, \prec)$ is a partially ordered structure. Then the following are equivalent:*

(a) *$D(\mathcal{P}) \leq 2$.*
(b) *The complete lattice embedding of $\mathcal{P}$ has a planar Hasse diagram.*

**Remark D.2.** *MacNeille (1937) and Birkhoff (1967, Chapter 5) introduced the construction of complete lattice embeddings for any partial order. Although it is difficult in practice to compute the complete lattice embedding for a partially ordered structure (MacNeille, 1937), Thm. D.1 can still provide an empirical characterization of the class of structures that can be efficiently represented. According to Euler's formula, the average degree of a vertex in a planar graph cannot exceed 6 (West, 2018, §6.1.23), which intuitively forces the partially ordered structures that can be represented as an intersection of 2 totally ordered structures to be sparse enough—thus to have planar complete lattice embeddings.*

*Fortunately, this is often the case in natural language. Such phenomenon is closely related to what is termed **valency** by Tesnière (1959, Part 1, Book D). The number of actants (i.e., arguments) needed to implement the function of a word is a property of the word itself—a constant that does not change with the context (cf. **categories**[9] in categorial grammars (Adjukiewicz, 1935; Bar-Hillel, 1953; Steedman, 1987)). In natural language, the valency of a word is often a small constant. For instance, Steedman (2000, Chapter 3, fn. 10 and Chapter 8, p. 212) observes that the highest valency in the Dutch and English lexicon can be regarded as bounded by $4$.*

---

[9]E.g., the English word "*give*" may have the category (VP/NP)/NP, meaning that it needs two NP categories to the right to form a VP. An example is the verb phrase "*give me an apple*", in which "*me*" and "*an apple*" are noun phrases.

We refer interested readers to MacNeille (1937) and Birkhoff (1967, Chapter 5) for the construction of complete lattice embeddings. Here, we give a weaker but more practical efficiency guarantee, based on a method to construct large partially ordered structures from smaller partially ordered structures.

**Definition D.3** (Series-Parallel Partial Orders; Valdes et al., 1979)**.** *A partially ordered structure is series-parallel if it satisfies the following inductive definition:*

(a) *A single-vertex structure with no edges is series-parallel;*

(b) *If partially ordered structures $\mathcal{P}_1 = (V_1, E_1, \prec)$ and $\mathcal{P}_2 = (V_2, E_2, \prec)$ are series-parallel, so is the partially ordered structures constructed by* either *of the following operations:*

   i. *Parallel composition:*
   $\mathcal{P}_\mathrm{p} = (V_1 \cup V_2, E_1 \cup E_2, \prec).$

   ii. *Series composition:*
   $\mathcal{P}_\mathrm{s} = (V_1 \cup V_2, E_1 \cup E_2 \cup (\mathcal{M}_1 \times \mathcal{N}_2), \prec),$ *where $\mathcal{M}_1$ is the set of sinks of $\mathcal{P}_1$ and $\mathcal{N}_2$ the set of sources of $\mathcal{P}_2$.*[10]

**Theorem D.4** (Series-parallel partially ordered structures are 2-dimensional; Valdes et al., 1979)**.** *The dimension of series-parallel partially ordered structures is at most 2.*

Thm. D.4 provides the guarantee that many structures in natural language processing can be represented as the intersection of 2 totally ordered structures. Since most structures of interest in NLP, such as trees and forests (thereby alignments and set partitioning), can be subsumed under series-parallel partially ordered structures, therefore have an order dimension of at most 2.

**Proposition D.5** (Trees are 2-dimensional; Lawler, 1978)**.** *Directed tree partially ordered structures are series-parallel. The order dimension of tree structures is at most 2.*

**Proposition D.6** (Forests are 2-dimensional)**.** *Forests are series-parallel. The order dimension of forest structures is at most 2.*

*Proof.* Forests are parallel compositions of trees. Thus, the proposition holds. ∎

## E  Efficient Algorithm for ⊕-Aggregation

### E.1  Correctness of Algorithm 1

---
**Algorithm 1** Computing $G_1$ when $K = 2$.

---
1: **procedure** COMPUTE-$G_1(f_{\boldsymbol{\theta}}^{(1)}, f_{\boldsymbol{\theta}}^{(2)}, V)$
2:    $U \leftarrow \texttt{sort}\left(V, \texttt{key} = f_{\boldsymbol{\theta}}^{(1)} - f_{\boldsymbol{\theta}}^{(2)}\right)$
3:    $G_1, s_1 \leftarrow \mathbf{0}, \mathbf{0} \triangleright \mathbf{0}$ *is the zero element of* ⊕
4:    **for** $n = 1$ **up to** $N$ :
5:       $q_1 = f_{\boldsymbol{\theta}}^{(1)}(U_n) + s_1 \triangleright q_1 = G_1(U_n)$
6:       $G_1 \oplus= q_1$
7:       $s_1 \oplus= -f_{\boldsymbol{\theta}}^{(1)}(U_n)$
8:    **return** $G_1$

---

**Proposition E.1.** *In Algorithm 1, $G_1 = \bigoplus_{x \in V} \bigoplus_{y \in \mathcal{S}_1(x)} \left( f_{\boldsymbol{\theta}}^{(1)}(x) - f_{\boldsymbol{\theta}}^{(1)}(y) \right).$*

*Proof.* By induction, we show that upon finishing step $n$, $s_1 = \bigoplus_{y \in \mathcal{S}_1(U_{n+1})} -f_{\boldsymbol{\theta}}^{(1)}(y)$, $G_1 = \bigoplus_{x \in \{U_1, \cdots, U_n\}} \bigoplus_{y \in \mathcal{S}_1(x)} \left( f_{\boldsymbol{\theta}}^{(1)}(x) - f_{\boldsymbol{\theta}}^{(1)}(y) \right)$. First, $\mathcal{S}_1(U_n) = \{U_1, \cdots, U_{n-1}\}$ holds as discussed in §4.3. When $n = 1$, we have $s_1 = -f_{\boldsymbol{\theta}}^{(1)}(U_1)$, $G_1 = \mathbf{0} = \bigoplus_{x \in \{U_1\}} \bigoplus_{y \in \mathcal{S}_1(x)} \left( f_{\boldsymbol{\theta}}^{(1)}(x) - f_{\boldsymbol{\theta}}^{(1)}(y) \right),$

---
[10]Sources and sinks refer to the vertices without incoming arcs and without outgoing arcs, respectively.

since $\mathcal{S}_1(U_1) = \varnothing$. Assume that our statements hold for $n = j$, when $n = j + 1$, it is straightforward that $s_1 = \bigoplus_{y \in \mathcal{S}_1(U_{j+2})} -f_{\boldsymbol{\theta}}^{(1)}(y)$. For $G_1$, we have

$$G_1 = \bigoplus_{x \in \{U_1, \cdots, U_j\}} \bigoplus_{y \in \mathcal{S}_1(x)} \left( f_{\boldsymbol{\theta}}^{(1)}(x) - f_{\boldsymbol{\theta}}^{(1)}(y) \right) \oplus \left( f_{\boldsymbol{\theta}}^{(1)}(U_{j+1}) + \bigoplus_{y \in \mathcal{S}_1(U_{j+1})} -f_{\boldsymbol{\theta}}^{(1)}(y) \right) \tag{7a}$$

$$= \bigoplus_{x \in \{U_1, \cdots, U_j\}} \bigoplus_{y \in \mathcal{S}_1(x)} \left( f_{\boldsymbol{\theta}}^{(1)}(x) - f_{\boldsymbol{\theta}}^{(1)}(y) \right) \oplus \bigoplus_{y \in \mathcal{S}_1(U_{j+1})} \left( f_{\boldsymbol{\theta}}^{(1)}(U_{j+1}) - f_{\boldsymbol{\theta}}^{(1)}(y) \right) \tag{7b}$$

$$= \bigoplus_{x \in \{U_1, \cdots, U_{j+1}\}} \bigoplus_{y \in \mathcal{S}_1(x)} \left( f_{\boldsymbol{\theta}}^{(1)}(x) - f_{\boldsymbol{\theta}}^{(1)}(y) \right) \tag{7c}$$

Thus, the claims hold for $n = j + 1$, establishing the induction step. ∎

**Proposition E.2.** *Algorithm 1 runs in $\mathcal{O}(N)$ time and space. With parallel computing, Algorithm 1 runs in $\mathcal{O}(\log N)$ span.*

*Proof.* The sorting step in line 2 can be executed in $\mathcal{O}(N)$ time and space. The for loop in lines 4 to 7 runs in $\mathcal{O}(N)$ time and space. In total, Algorithm 1 runs in $\mathcal{O}(N)$ time and space. Computing $s_1$ in each step is a prefix-sum of $-f_{\boldsymbol{\theta}}^{(1)}(U_n)$, which can be done in $\mathcal{O}(\log N)$ span with parallel computing. $q_1, G_1$ in each step can be computed in $\mathcal{O}(1)$ in parallel following the computation of all $s_1$. Thus, the total span of Algorithm 1 is $\mathcal{O}(\log N)$. ∎

### E.2  Order Dimension $K > 2$

Finding all $y \in V$ such that $x \prec y$ in a partial order for a given $x \in V$ requires efficiently finding all $y$ that satisfy $\bigwedge_{k \in [K]} (f_{\boldsymbol{\theta}}^{(k)}(x) < f_{\boldsymbol{\theta}}^{(k)}(y))$. We remark that this problem bears a resemblance to **orthogonal range searching** in a $K$-dimensional space (Berg et al., 2008, Chapter 5), i.e., for a given $x$, we aim to find all $y$ such that $(f_{\boldsymbol{\theta}}^{(1)}(y), f_{\boldsymbol{\theta}}^{(2)}(y), \cdots, f_{\boldsymbol{\theta}}^{(K)}(y))$ is within the range $(f_{\boldsymbol{\theta}}^{(1)}(x), \infty) \times (f_{\boldsymbol{\theta}}^{(2)}(x), \infty) \times \cdots \times (f_{\boldsymbol{\theta}}^K(x), \infty)$. This problem can be naïvely solved in $\mathcal{O}(\log^{K-1} N + \ell)$ using a **range tree** (Bentley, 1979, 1980; Chazelle, 1988, 1990a,b), where $\ell$ is the cardinality of query results, as opposed to arc-factored models in which solving the same problem takes $\mathcal{O}(N)$ computations.

For $\oplus$-aggregation, a more efficient algorithm which makes use of $(K-1)$-dimensional range trees can be designed. In future work, we show that computing the complexity of $\oplus$-aggregation for *all $x \in V$* can be further reduced to $\mathcal{O}(KN \log^{K-2} N)$ by applying Fredman's (1976) trick which we used in Algorithm 1. Extending the notation in §4.3, the set of all vertices $V$ can be partitioned into $K$ subsets $\mathcal{S}_1(x), \cdots, \mathcal{S}_K(x)$ for each $x \in V$, where $\mathcal{S}_k(x) = \{y \mid y \in V \wedge F_{\boldsymbol{\theta}}(x, y) = f_{\boldsymbol{\theta}}^{(k)}(x) - f_{\boldsymbol{\theta}}^{(k)}(y)\}$. $\bigoplus_{y \in V} F_{\boldsymbol{\theta}}(x, y)$ can be decomposed into a $\oplus$-aggregation of $K$ terms.

$$G(x) \overset{\text{def}}{=} \bigoplus_{y \in V} F_{\boldsymbol{\theta}}(x, y) \tag{8a}$$

$$G(x) = \bigoplus_{k \in [K]} \underbrace{\left( \bigoplus_{y \in \mathcal{S}_k} F_{\boldsymbol{\theta}}(x, y) \right)}_{\overset{\text{def}}{=} G_k(x)} \tag{8b}$$

We leave to future work showing that computing each $G_k(x)$ takes $\mathcal{O}(\log^{K-2} N)$.

## F  Hyperparameter Settings

### F.1  Dependency Parsing

For pretrained language models, we use XLNet-large-cased[11] (Yang et al., 2019) for PTB, bert-base-chinese[12] for CTB, and bert-base-multilingual-cased[13] for UD. We set the dimension of POS tag embedding to 256 for all experiments. On top of concatenated pretrained representations

---

[11] https://huggingface.co/xlnet-large-cased
[12] https://huggingface.co/bert-base-chinese
[13] https://huggingface.co/bert-base-multilingual-cased

and POS embedding, we use a 3-layer BiLSTM (Hochreiter and Schmidhuber, 1997) with a hidden size of 768 for base-sized models (`bert-base-chinese` on CTB and `bert-multilingual-cased` on UD) and 1024 for large-sized models (`xlnet-large-cased` on PTB). We apply dropout with a rate of 0.33 to the concatenated embedding layer, between LSTM layers, and before the linear projection layer of the realizer. We employ AdamW (Loshchilov and Hutter, 2019) with a learning rate of $2\mathrm{e}-5$ for pretrained LMs and $1\mathrm{e}-4$ for POS embedding, BiLSTM, and linear projection during training. The gradient clipping threshold is set to $1.0$. The batch size for training is $32$. The number of training epochs is $50$.

### F.2 Coreference Resolution

We use `longformer-large-cased`[14] (Beltagy et al., 2020) as the pretrained encoder. We use the same hyperparameter settings as Kirstain et al. (2021). We use AdamW with a learning rate of $1\mathrm{e}-5$ for pretrained LM and $3\mathrm{e}-4$ for the linear projection during training, with 5600 linear warmup steps. Training documents are batched into batches with maximum 5000 tokens in total. The number of training epochs is 129.

## G Datasets

### G.1 Dependency Parsing

**Preprocessing.** We follow previous work (Kiperwasser and Goldberg, 2016; Dozat and Manning, 2017) to derive the dependency annotations from the treebank annotations using the Stanford Dependency converter v3.3.0 (de Marneffe and Manning, 2008). During evaluation, punctuations are omitted. Following Amini et al. (2023), we provide gold part-of-speech tags to the model during training and decoding.

**Splits.** The dataset splits are consistent with previous work. For PTB, we follow the standard split of Marcus et al. (1993), resulting in 39,832 sentences for training, 1,700 for development, and 2,416 for testing. For CTB, we follow the split of Zhang and Clark (2008), resulting in 16,091 sentences for training, 803 for development, and 1,910 for testing. For UD, we follow previous work (Zhang et al., 2020; Yang and Tu, 2022) and use the standard splits of the following corpora for experiments: BG-btb, CA-ancora, CS-pdt, DE-gsd, EN-ewt, ES-ancora, FR-gsd, IT-isdt, NL-alpino, NO-rrt, RO-rrt, RU-syntagrus.

**Licenses.** The PTB and CTB datasets are licensed under LDC User Agreement. The UD dataset is licensed under the Universal Dependencies License Agreement.

### G.2 Coreference Resolution

**Preprocessing.** We experiment on the CoNLL-2012 English shared task dataset (OntoNotes; Pradhan et al., 2012). We follow the preprocessing procedure of (Kirstain et al., 2021). During training and decoding, the speaker information is provided to the model.

**Splits.** The OntoNotes dataset contains 2,802 documents for training, 343 for validation, and 348 for testing. We use this official split following previous work (Lee et al., 2017; Kirstain et al., 2021).

**Licenses.** The OntoNotes dataset is licensed under LDC User Agreement.

## H Efficiency Evaluation

### H.1 Dependency Parsing

For efficiency evaluation, `BERT-large-cased`[15] is used as the pretrained encoder for our method with $K = 2$, hexatagger (Hexa; Amini et al., 2023), and biaffine model (`Biaff`). We use the English PTB test set and truncate or pad the input sentences to the control length. The results are averaged over 3 random runs on the same server with one Nvidia A100-80GB GPU. The other experimental settings are kept the same (i.e., the version of PyTorch and `transformers`, FP32 precision, batching).

### H.2 Coreference Resolution

---

[14]https://huggingface.co/allenai/longformer-large-4096
[15]https://huggingface.co/bert-large-cased
[16]https://huggingface.co/allenai/longformer-base-4096

| Doc length | Speed (doc/s) ↑ | | Memory (GB) ↓ | |
|---|---|---|---|---|
| | Ours ($K = 4$) | Kirstain et al. | Ours ($K = 4$) | Kirstain et al. |
| 512 | 72.5 | 35.7 | 7.3 | 7.4 |
| 1024 | 54.3 | 26.7 | 7.3 | 7.4 |
| 2048 | 33.8 | 15.9 | 9.4 | 9.5 |
| 4096 | 19.3 | 8.6 | 17.8 | 21.0 |
| overall | **82.8** | 41.9 | **7.3** | 7.4 |

Table 4: Comparison of speed and memory consumption on OntoNotes test set using Longformer-base[16] as pretrained encoder. Results are averaged over 3 random runs on the same server with one Nvidia A100-80GB GPU using BERT-large as encoder. We use a batch size of 32 documents.

We compare the efficiency of our order-theoretic method with baseline coreference resolution model. The full results are given in Tab. 4. On the OntoNotes coreference resolution benchmark, our method is twice as fast as Kirstain et al.'s (2021) model while using less memory, especially on long documents. It is worth noting that Kirstain et al. (2021) has already performed aggressive optimization in both the speed and memory footprint of coreference modeling. I.e., they abandon the computation for textual span representations and entity-pair representations, and use biaffine scorers to compute coreference scores.

# I Additional Experimental Results

## I.1 Dependency Parsing

We report additional experimental results on the UD dependency parsing dataset in Tab. 5. On average, our model has state-of-the-art performance and outperforms all other baseline models on 5 languages.

| | bg | ca | cs | de | en | es | fr | it | nl | no | ro | ru | Avg. |
|---|---|---|---|---|---|---|---|---|---|---|---|---|---|
| Zhang et al. (2020) | 90.77 | 91.29 | 91.54 | 80.46 | 87.32 | 90.86 | 87.96 | 91.91 | 88.62 | 91.02 | 86.90 | 93.33 | 89.33 |
| Wang and Tu (2020) | 90.53 | 92.83 | 92.12 | 81.73 | 89.72 | 92.07 | 88.53 | 92.78 | 90.19 | 91.88 | 85.88 | 92.67 | 90.07 |
| +BERT_multilingual | | | | | | | | | | | | | |
| Wang and Tu (2020) | 91.30 | 93.60 | 92.09 | 82.00 | 90.75 | 92.62 | 89.32 | 93.66 | 91.21 | 91.74 | 86.40 | 92.61 | 90.61 |
| Dozat and Manning (2017) | 90.30 | **94.49** | 92.65 | **85.98** | 91.13 | 93.78 | **91.77** | 94.72 | 91.04 | 94.21 | 87.24 | **94.53** | 91.82 |
| Yang and Tu (2022) | 91.10 | 94.46 | 92.57 | 85.87 | 91.32 | **93.84** | 91.69 | **94.78** | 91.65 | **94.28** | 87.48 | 94.45 | 91.96 |
| Amini et al. (2023) | 92.87 | 93.79 | 92.82 | 85.18 | 90.85 | 93.17 | 91.50 | 94.72 | 91.89 | 93.95 | 87.54 | 94.03 | 91.86 |
| ours ($K = 2$) | 92.81 | 93.26 | 92.52 | 83.33 | 90.38 | 92.55 | 89.83 | 93.82 | 91.29 | 93.61 | 87.40 | 94.10 | 91.24 |
| ours ($K = 4$) | **93.82** | 94.23 | **93.03** | 84.68 | **91.40** | 93.62 | 90.95 | 94.59 | **92.58** | 94.22 | **88.45** | 94.40 | **92.16** |

Table 5: LAS scores on the test sets of 12 languages in UD. Our method with an order dimension of $K = 4$ achieves competitive performance in all languages, being state-of-the-art on 5 languages and on average.

# J Qualitative Examples

We present some qualitative examples from the PTB development set and one non-projective example using our method with a 2-dimensional realizer, with their ground truth annotations on the right in Figures 4–9. For a more intuitive and compact exhibition, we plot the 2 total orders output by our model in a 2-dimensional plane. Each axis corresponds to one of the 2 orders. The relation $x \prec y$ encoded by $\bigwedge_{k \in \{1,2\}} f_{\boldsymbol{\theta}}^{(k)}(x) < f_{\boldsymbol{\theta}}^{(k)}(y)$ is equivalent to $x$ being located *below and to the left* of $y$.

Tokens in $V^r$ and $V^b$ are represented by ✗ and ●, respectively. The line segments between ✗ and ● are the extracted dependency relations. In each of the plots, every ✗ (token in $V^r$) except for the root is connected to a ● (token in $V^b$), which indicates ✗ is the modifier of ●. The roots (*about, moving, ready, had, adds, bought* represented by ✗) are not connected to any other word.

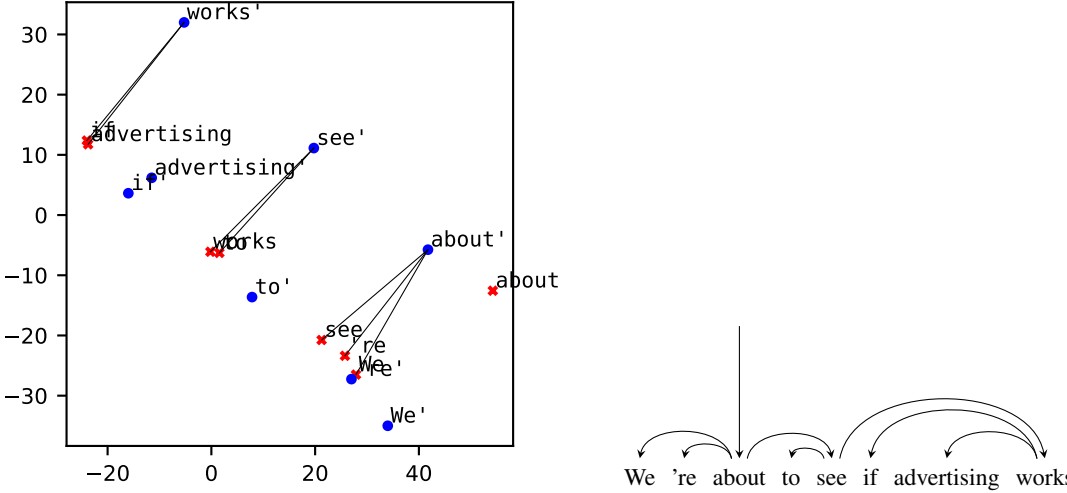

Figure 4: *We 're about to see if advertising works*

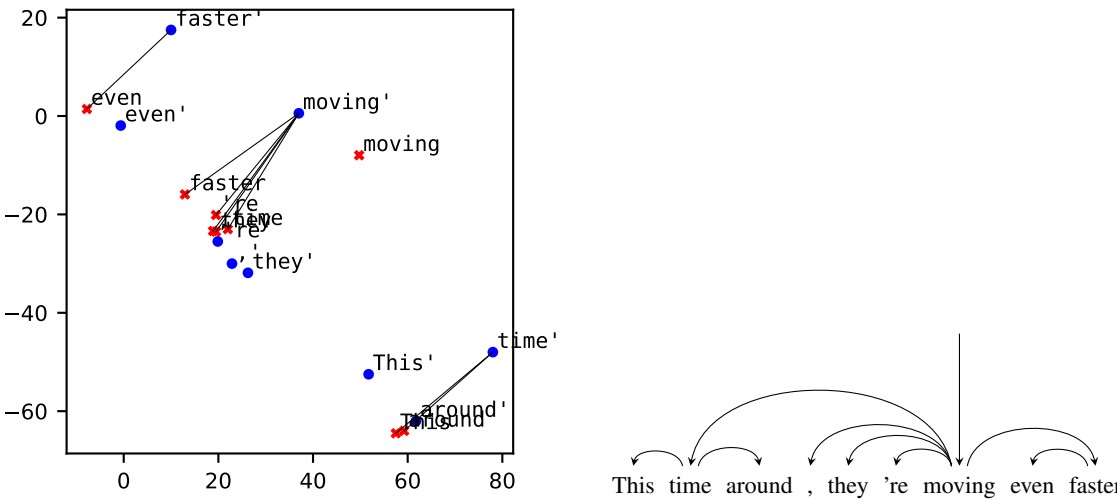

Figure 5: *This time around , they 're moving even faster*

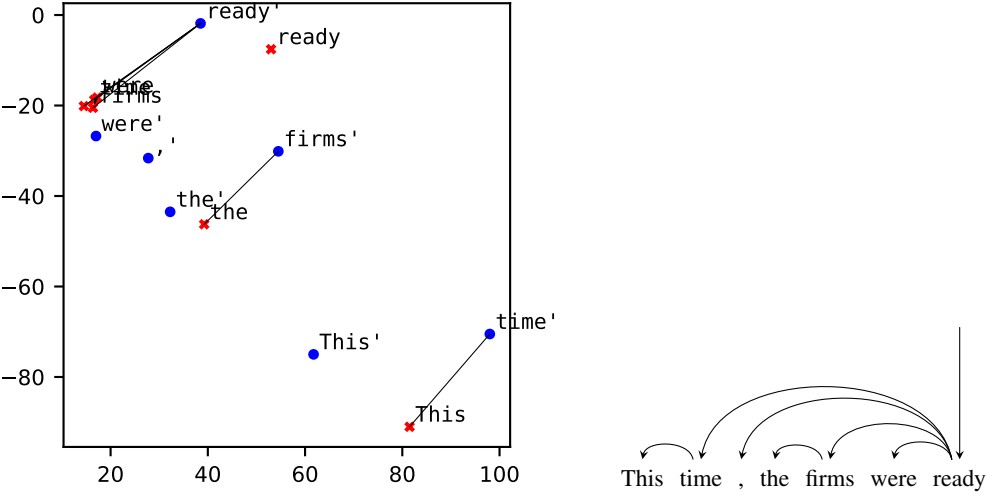

Figure 6: *This time , the firms were ready*

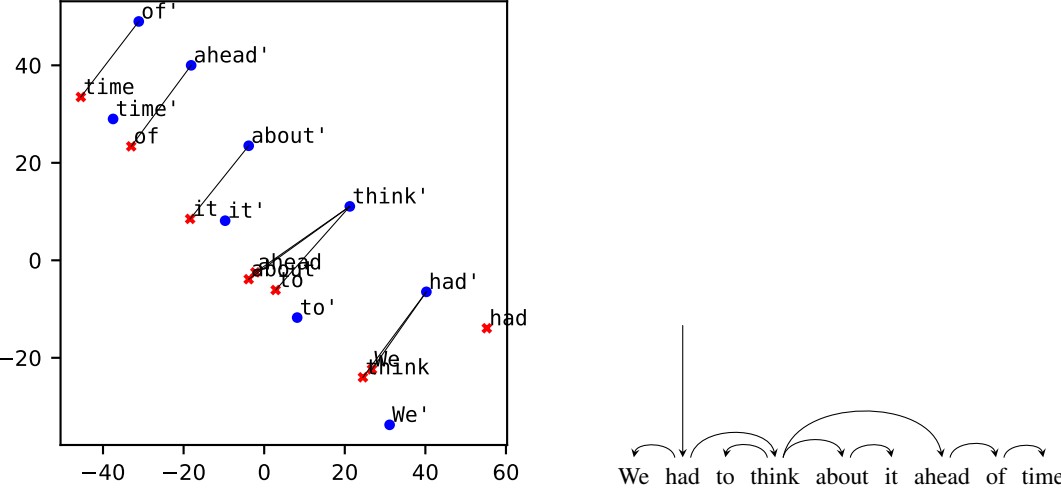

Figure 7: *We had to think about it ahead of time*

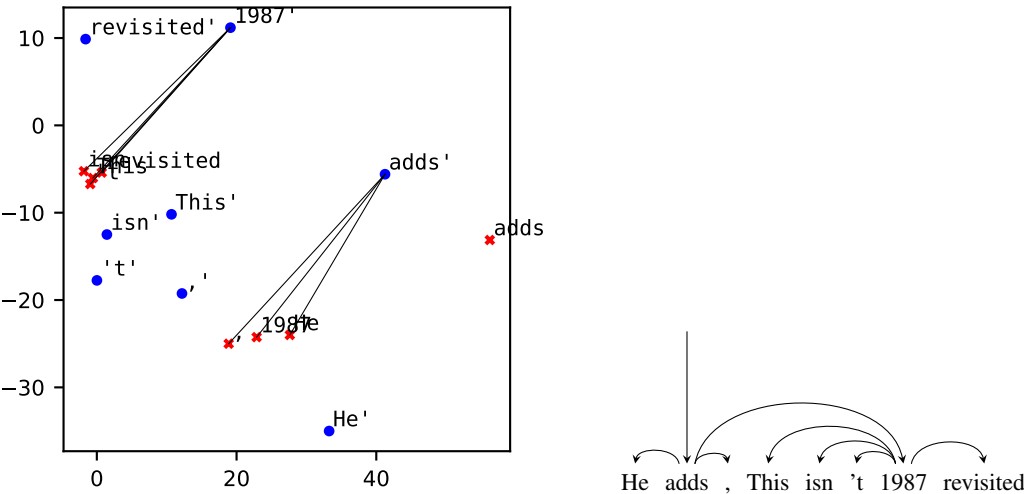

Figure 8: *He adds , " This isn 't 1987 revisited "*

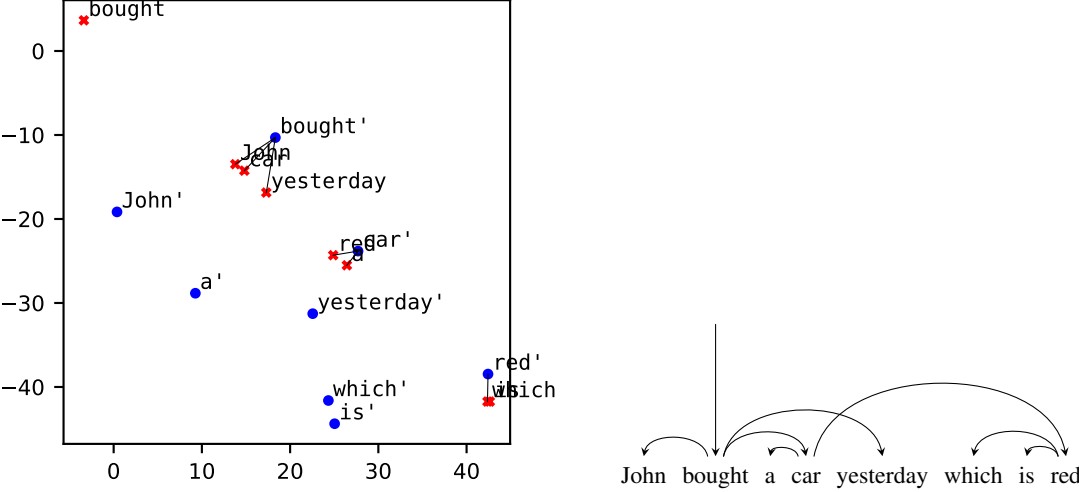

Figure 9: A sentence with a non-projective dependency structure: *John bought a car yesterday which is red*