# OpenReview forum: "Linear-Time Modeling of Linguistic Structure: An Order-Theoretic Perspective"
_EMNLP/2023/Conference — EMNLP 2023 Main_

### Official Review · Reviewer_uTtf · 2023-07-26

**Soundness:** 5

**Excitement:**

5: Transformative: This paper is likely to change its subfield or computational linguistics broadly. It should be considered for a best paper award. This paper changes the current understanding of some phenomenon, shows a widely held practice to be erroneous in someway, enables a promising direction of research for a (broad or narrow) topic, or creates an exciting new technique.

**Missing References:**

I found the framing in Section 2 somewhat narrow. Some linearization schemes are cited, and then lack of coverage is mentioned as a drawback. But this is because the particular approaches that are cite are those that prioritize compactness over coverage (they have a constant time number of tags, but are limited to projective trees/continuous constituency trees). There are other linearization approaches with the converse priority which are not cited, e.g. (Strzyz et al., 2020, strzyz-etal-2020-bracketing) or some of the encodings in the papers cited in Section 7.3, it seems like both kinds should be mentioned, especially given the fact that the coverage vs. compactness tradeoff is explicitly mentioned in said Section 2.

Another paper that I believe should be cited is (Shen et al., 2018, shen-etal-2018-straight): although their method does not seem related to this one, it has in common that a parse tree is represented via a sequence of real numbers.

**Paper Topic And Main Contributions:**

This paper presents a new approach to structured prediction problems in NLP whose output is in the form of a directed graph. The method is based on representing the directed graph as a partial order on a "token-split" set (a set where each input token is included twice, once as a potential source of outgoing arcs and once as a target of incoming arcs). This partial order can, in turn, be represented as the intersection of k total orders (in particular, for the cases of trees and forests, it is shown by referring to order-theoretic results that k=2 is enough). To use this in practice, a neural model is trained to generate 2k real numbers per token. These numbers uniquely determine the k total orders for the token-split set, and thus can be used to recover the tree or forest. Experiments are performed on dependency parsing and coreference resolution, showing competitive results and remarkable efficiency.

**Questions For The Authors:**

As mentioned above: if there is a theoretical guarantee that k=2 is enough for trees, why does k=4 perform better? And if k=4 performs better (regardless of the reason), why not make it even larger?

**Reasons To Accept:**

The contribution of the paper is highly original, proposing a generic method that can be applied to various structured prediction problems and, as far as I know, differs from the existing approaches in the literature.

The method works well empirically, as shown in the experiments. This is not only in terms of accuracy, but it is also very efficient.

The paper establishes interesting connections to parts of the discrete mathematics literature that are not well-known by the NLP community, while being clearly written and understandable without prior exposure to that literature.

**Reasons To Reject:**

Dependency parsing experiments are done with gold PoS tags. This makes the comparison in Table 1 misleading, as most of the parsers included there use predicted PoS tags. For example, from a quick look at Table 1 it would seam that the k=4 system is tied with Mrini et al., but Mrini et al. uses predicted PoS tags. The advantage gained from using gold with respect to predicted PoS tags is well attested in the literature (Yasunaga et al., 2018, yasunaga-etal-2018-robust; Anderson and Gómez-Rodríguez, 2020, anderson-gomez-rodriguez-2020-frailty)

One aspect is unclear: according to Propositions 4.16 and 4.17, trees and forests can be represented as intersections of only two posets. Therefore, k=2 should be enough for full-coverage dependency parsing. However, experiments are made with k=2 and k=4, with k=4 obtaining better results. Why does this happen? If k=2 is sufficient, one wouldn't expect introducing extra complexity into the model should be beneficial.

**Reproducibility:**

4: Could mostly reproduce the results, but there may be some variation because of sample variance or minor variations in their interpretation of the protocol or method.

**Reviewer Confidence:**

4: Quite sure. I tried to check the important points carefully. It's unlikely, though conceivable, that I missed something that should affect my ratings.

**Typos Grammar Style And Presentation Improvements:**

In Table 2, the exact meaning of #token is unclear: I suppose it refers to sentence length, but what exactly: the set of sentences of exactly that length, the set of sentences with length <= that, buckets [1,32], [33, 64]...?

line 273: missing "such" in "such that"

line 486: missing s in "linear projections"

line 676: "empirical experiments" is redundant

line 1193: redundancy "set... is set to"

Since the method supports non-projective trees, it would be nice to include a non-projective tree example in Appendix J.

In the experimental section, the UD treebanks used and the hardware used in the efficiency measures should be moved to the main text; apart from being clear about the use of gold PoS tags and its impact in comparisons as mentioned above.

---

> ### Author Rebuttal · Authors · 2023-08-25
>
> We appreciate the reviewer for the valuable feedback, which we believe will greatly improve our paper. We address the concerns in the review and propose revisions to our paper regarding them below.
>
> 1. Regarding the suggested references in Sec. 2 and Sec. 7:
>     *  We will extend Sec. 2 with more thorough literature by including the suggested references on linearizing dependency trees (Strzyz et al., 2020 [5]).
> In Sec. 7, we will further discuss the line of work on the multiplanarity of dependency trees (Yli-Jyrä, 2003 [9]; Gómez-Rodríguez & Nivre, 2010 [3]; Strzyz et al., 2020 [5]; inter alia) and its relevance to our work.
>     *  We will add a new subsection in Sec. 7 discussing the usage of “learning-to-rank” and the mapping between real number sequences and structures (Shen et al., 2018ab [6,7]).
>
>
> 2. Regarding the experiment on PTB with gold POS tags:
>     *  In our experiment, we reproduced (Mrini et al., 2020) using the released script which might not be in accord with the paper in some detail. Specifically, https://github.com/KhalilMrini/LAL-Parser/blob/master/src_joint/dep_reader.py#L67 takes the 5th column from a CoNLL-U formatted PTB as input, which corresponds to the language-specific POS tag (XPOS) [1] rather than predicted POS tag. We used the **same [PTB data](https://github.com/KhalilMrini/LAL-Parser/tree/master/data)** as the repo of (Mrini et al., 2020) and the **same columns** as model inputs in the experiments in Table 1, and achieved comparable or better performance.
> When not using any POS tags on PTB, our model gets 95.5 LAS, 97.1 UAS using $k=2$, 95.7 LAS, 97.3 UAS using $k=4$, which is also comparable with the SOTA without POS tags. In the revised version, we will report dependency parsing results with both gold tags and no tags and further clarify the experimental settings.
>     *  In addition, we will also cite the suggested references (Yasunaga et al., 2018 [8]; Anderson and Gómez-Rodríguez, 2020 [1]) and include discussions on the possible impact of using or not using gold POS tags.
>
> 3. Regarding order dimension $ k $:
>     *  Consider modeling a tree structure with $N$ nodes with $N-1$ arcs using partial order, which implies $ N - 1 $ constraints of the form $ x \prec y’$ and $ N^2 -2N+1 $ constraints of $ x \nprec y’ $.
> From a theoretical perspective, $k = 2$ is sufficient to represent such a structure as shown in Sec. 4. I.e., there always exist 2 total orders whose intersection satisfies the aforementioned $N(N-1)$ constraints. However, it might not be the easiest to learn with a limited number of training data.
> &nbsp;
> Increasing $k$ beyond 2 can **facilitate the satisfaction of constraints**, especially of the form $x \nprec y’$---since there are more constraints in this form. It allows more possibilities for $\bigvee_{i\in [k]} f_i(x) \ge f_i(y’)$ (i.e., more choices of $i$ to satisfy the expression). We conclude that this accounts for the increase in performance when increasing $k$ from 2 to 4.
> In our preliminary experiments, we found using $k > 4$ does *not* bring improvements to dependency parsing, and only brings *marginal* improvements to coreference resolution, from 79.2 F1 ($k=4$) to 79.6 F1 ($k=8$). Thus, we did not choose $k$ that is greater than 4.
>     *  In the revised version, we will include a discussion on how both theoretical and practical factors affect the performance of a parser. Furthermore, we will extend the “Hardness of Learning” subsection in the Limitations by including the above example.
>
> 4. Regarding suggestions on improvements for presentation
>     *  We have carefully corrected the typos.
>     *  We will restructure the experimental settings by moving important details to the main text to improve clarity. The first column of Table 2 refers to the set of sentences of the lengths.
>     *  We will include non-projective sentences from the **UD** dataset as qualitative examples in App. J, as the versions of PTB and CTB we use for evaluation are projective.
>
>
>
> [1] https://universaldependencies.org/format.html
> [2] Mark Anderson and Carlos Gómez-Rodríguez. 2020. On the Frailty of Universal POS Tags for Neural UD Parsers.
> [3] Carlos Gómez-Rodríguez and Joakim Nivre. 2010. A Transition-Based Parser for 2-Planar Dependency Structures.
> [4] Carlos Gómez-Rodríguez and Joakim Nivre. 2013. Divisible Transition Systems and Multiplanar Dependency Parsing.
> [5] Michalina Strzyz, David Vilares, and Carlos Gómez-Rodríguez. 2020. Bracketing Encodings for 2-Planar Dependency Parsing.
> [6] Yikang Shen, Zhouhan Lin, Athul Paul Jacob, Alessandro Sordoni, Aaron Courville, and Yoshua Bengio. 2018. Straight to the Tree: Constituency Parsing with Neural Syntactic Distance.
> [7] Yikang Shen, Zhouhan Lin, Chin-wei Huang, and Aaron Courville. 2018. Neural Language Modeling by Jointly Learning Syntax and Lexicon.
> [8] Michihiro Yasunaga, Jungo Kasai, and Dragomir Radev. 2018. Robust Multilingual Part-of-Speech Tagging via Adversarial Training.
> [9] Anssi Mikael Yli-Jyrä. 2003. Multiplanarity - a model for dependency structures in treebanks.

---

### Official Review · Reviewer_esuW · 2023-07-28

**Soundness:** 5

**Excitement:**

5: Transformative: This paper is likely to change its subfield or computational linguistics broadly. It should be considered for a best paper award. This paper changes the current understanding of some phenomenon, shows a widely held practice to be erroneous in someway, enables a promising direction of research for a (broad or narrow) topic, or creates an exciting new technique.

**Missing References:**

In section 5.2, since the proposed loss function bears resemblance to the 'pairwise learning-to-rank loss', it would be beneficial to include some discussions in the related work section. Additionally, I would like to mention a previous work that utilized the 'learning-to-rank' loss for constituency parsing (Straight to the Tree: Constituency Parsing with Neural Syntactic Distance, https://aclanthology.org/P18-1108.pdf).






**Paper Topic And Main Contributions:**

This research aims to address the challenges of time and space complexity in NLP structured prediction problems. The authors propose an elegant linear-time-and-space algorithm that utilizes the partial order of a string. They demonstrate that by intersecting multiple total orders, the desired structures of interest, including important NLP structures like set partitions, trees, and alignments, can be recovered.

Additionally, the authors use neural networks to parameterize posets/total orders and introduce a "learning-to-rank"-like loss function for training. This approach leads to competitive or state-of-the-art performance in two classic NLP structured prediction tasks: dependency parsing and coreference resolution.






**Reasons To Accept:**

- This paper demonstrates excellent writing, as seen in Section 3 Motivating Example, which immediately conveys the key idea to the reader.

- The paper is built on a solid theoretical foundation and presents an elegant approach.

- The proposed model has wide applicability across various NLP structured prediction tasks. The authors thoroughly evaluate their elegant model in two significant tasks: dependency parsing and coreference resolution, achieving competitive or even state-of-the-art performance.






**Reasons To Reject:**

None that I can see.

**Reproducibility:**

4: Could mostly reproduce the results, but there may be some variation because of sample variance or minor variations in their interpretation of the protocol or method.

**Reviewer Confidence:**

4: Quite sure. I tried to check the important points carefully. It's unlikely, though conceivable, that I missed something that should affect my ratings.

---

> ### Author Rebuttal · Authors · 2023-08-26
>
> We thank the reviewer for the valuable feedback. We will carefully address the concerns in the revised manuscript.
> Specifically, regarding the suggested references, we will include (Shen et al., 2018ab [6,7]) in the related work section and further discuss the usage of “learning-to-rank” style losses for structured prediction.

---

### Official Review · Reviewer_JosL · 2023-08-04

**Soundness:** 5

**Excitement:**

5: Transformative: This paper is likely to change its subfield or computational linguistics broadly. It should be considered for a best paper award. This paper changes the current understanding of some phenomenon, shows a widely held practice to be erroneous in someway, enables a promising direction of research for a (broad or narrow) topic, or creates an exciting new technique.

**Paper Topic And Main Contributions:**

They explore a different paradigm for building structured prediction systems. Rather than directly learning a quadratic mapping where the domain is the spans of an input sentence, they leverage results from the mathematical literature about partial and total orders to express structured prediction problems using a more tractable representation. First, they observe that many structured prediction problems can be expressed as a partial ordering of vertices of a bipartite graph (a so-called "token-split structure"), where each token has one vertex in each partition. Then, they leverage literature that shows that partial orders can always be represented as an intersection of total orders, and observe that for many structure prediction tasks, the relevant partial orders can be represented as an intersection of 2 total orders.

From here, they realize that this means we can train structured prediction models by having a neural network learn total orders whose intersection is the partial order corresponding to the structure. The results (on dependency parsing and coref resolution) are strongly competitive with state-of-the-art approaches. However, the representations are linear (rather than quadratic or worse) in the sentence length, and thus asymptotically more efficient.

**Reasons To Accept:**

This paper is quite convincing, and its approach is nothing like anything I've seen before in the structured prediction literature.

**Reasons To Reject:**

I see no red flags here.

**Reproducibility:**

4: Could mostly reproduce the results, but there may be some variation because of sample variance or minor variations in their interpretation of the protocol or method.

**Reviewer Confidence:**

3: Pretty sure, but there's a chance I missed something. Although I have a good feel for this area in general, I did not carefully check the paper's details, e.g., the math, experimental design, or novelty.

**Typos Grammar Style And Presentation Improvements:**

Sec 2: Not sure you strictly need a reference for the fact that the space of real numbers is infinite.
Footnote 1: A homogeneous relation on a set X is a binary relation over element pairs of X?
Def 4.5: I feel like the statement of the definition could be improved.
Def 4.9: I wouldn't put "The intersection of tosets is a poset" as part of definition. It doesn't seem definitional.
Def 4.11: "a set of toset" => "a set of tosets"

---

> ### Author Rebuttal · Authors · 2023-08-26
>
> We appreciate the reviewer for the valuable feedback.
> We have corrected the typos. In the revised manuscript, we will improve the clarity and rigor of the definitions of concepts.

---

### Meta-Review · Area_Chair_u1g1 · 2023-09-08

**Recommendation:** 5

**Metareview:**

This paper presents an original, efficient, and state-of-the-art (SOTA) paradigm for computing structured representations of natural language inputs. The work uses the concept of a partial order of a string to represent the relations in a bipartite graph. In particular, the authors propose that by generating several total orders of the string and combining them, they can recover the desired relations in the output graph. The study initially introduces a robust and engaging theoretical demonstration of the foundational aspects of the approach. Subsequently, it implements a functional neural model that achieves highly impressive results in the domains of dependency parsing and coreference resolution.

There is unanimous consensus among the reviewers regarding the acceptance of this paper. They clearly acknowledge its novelty and significance within the field. It's an elegant approach with a strong theoretical foundation and wide applicability across various NLP tasks and structured prediction problems, making it a standout contribution.

The reviewers suggest minor changes, including providing a more comprehensive overview of the related previous work, along with other minor technical suggestions, which can be easily addressed in the camera-ready version of the paper.

---

### Decision · Program_Chairs · 2023-10-07

**Decision:**

Accept-Main

**Comment:**

This paper presents an original, efficient, and state-of-the-art (SOTA) paradigm for computing structured representations of natural language inputs. The work uses the concept of a partial order of a string to represent the relations in a bipartite graph. In particular, the authors propose that by generating several total orders of the string and combining them, they can recover the desired relations in the output graph. The study initially introduces a robust and engaging theoretical demonstration of the foundational aspects of the approach. Subsequently, it implements a functional neural model that achieves highly impressive results in the domains of dependency parsing and coreference resolution.

There is unanimous consensus among the reviewers regarding the acceptance of this paper. They clearly acknowledge its novelty and significance within the field. It's an elegant approach with a strong theoretical foundation and wide applicability across various NLP tasks and structured prediction problems, making it a standout contribution.

The reviewers suggest minor changes, including providing a more comprehensive overview of the related previous work, along with other minor technical suggestions, which can be easily addressed in the camera-ready version of the paper.